# Thrombosis in Multisystem Inflammatory Syndrome Associated with COVID-19 in Children: Retrospective Cohort Study Analysis and Review of the Literature

**DOI:** 10.3390/biomedicines11082206

**Published:** 2023-08-06

**Authors:** Liudmila V. Bregel, Olesya S. Efremova, Kirill Y. Kostyunin, Natalya Y. Rudenko, Yury A. Kozlov, Vadim V. Albot, Natalya А. Knyzeva, Olga V. Tolmacheva, Svetlana V. Ovanesyan, Alexander O. Barakin, Ki O. Pak, Liudmila V. Belousova, Tatyana S. Korinets, Mikhail M. Kostik

**Affiliations:** 1Department of Pediatrics, Irkutsk State Medical Academy of Postgraduate Education, Branch of Russian Medical Academy of Continuous Professional Education, 664049 Irkutsk, Russia; 2Irkutsk Regional Children’s Hospital, 664022 Irkutsk, Russia; 3Irkutsk Regional Diagnostic Centre, Department of Clinical Pathomorpholigy, 664047 Irkutsk, Russia; kostjunin@gmail.com; 4Pathology Department, Irkutsk State Medical University, 664003 Irkutsk, Russia; 5Hospital Pediatry, Saint Petersburg State Pediatric Medical University, 194100 Saint Petersburg, Russia

**Keywords:** SARS-CoV-2, coronavirus, COVID-19, multisystem inflammatory syndrome in children, MIS-C, thrombosis

## Abstract

**Background:** The causative agent of the new coronavirus infection SARS-CoV-2 has unique properties causing hyperinflammatory syndrome and cytokine storm, as well as widespread endotheliitis and thrombotic microangiopathy, initially detected in the lungs of adult patients who died from a severe form of the disease. Venous and arterial thrombosis in adults were identified as common causes of severe complications and deaths in new coronavirus infections. There are very few reports of thrombotic events in children with COVID-19 in the literature. **Methods:** We conducted a retrospective analysis of the histories of 60 patients in the Irkutsk Regional Children’s Clinical Hospital from November 2020 to November 2022 with a MIS-C diagnosis established according to WHO criteria, of which 8 (13.3%) were diagnosed with venous and/or arterial thrombosis, confirmed by laboratory and ultrasound and/or X-ray methods. **Results:** The average age of children with thrombosis (Me) was 7.5 years (min 4 months, max 17 years), with a M:F ratio of 3.0. Venous thrombosis was detected in six of the eight patients, including in the deep veins of the lower extremities in four. Pulmonary embolism occurred in two (one of them was fatal), and cerebral venous sinus thrombosis and thrombosis of the branches of the upper and lower vena cava were found in one patient. Extensive bilateral stroke due to thrombosis of the large cerebral arteries occurred in two patients, including one in combination with distal gangrene. Secondary thrombotic renal microangiopathy took place in three of the eight patients. Among these three, atypical HUS was diagnosed in one case. Multiple thrombosis involving the venous and arterial bed was detected in four of the eight patients. High levels of D-dimer, thrombocytopenia, increased NT-proBNP, cerebral coma, and aseptic meningitis were the events most often associated with thrombosis. All patients received immunomodulatory therapy (immunoglobulin, dexamethasone/methylprednisolone), pathogenetic therapy for multiorgan failure, anticoagulant therapy with heparin/LMWH, and acetylsalicylic acid. Biologics were used in two patients. **Conclusions:** The main predictors of thrombosis in children with MIS-C were increased D-dimer, thrombocytopenia, hospitalization in the ICU, and noncardiogenic pulmonary edema. Thrombosis of the deep veins of the lower extremities, large cerebral arteries, and secondary thrombotic microangiopathy was common. There was a single death (12.5% of the eight patients), associated with PE.

## 1. Introduction

SARS-CoV-2 infection, which most commonly affects the upper respiratory tract, induces the development of prothrombotic status in some patients. This is due to the unique properties of the pathogen SARS-CoV-2, and is clinically manifested especially by microthrombosis, observed both in vivo and in autopsy of patients who died of COVID-19. These thromboses are associated with microangiopathy provoked by hyperinflammatory status and cytokine storm, and were initially found in the lungs of adult patients in the severe course of the active phase of COVID-19 [1,2,3,4].

Thrombotic complications of COVID-19 in adults are widely described in the literature, including deep vein thrombosis of the extremities, pulmonary embolism and catheter-associated venous thrombosis, microangiopathic renal damage, and arterial and venous thromboembolic complications, clinically manifested as acute ischemic stroke [5,6,7].

Thrombotic complications in adults are a major risk factor for fatal outcomes, almost doubling their probability (23% in patients with thrombotic events vs. 13% in patients without documented thrombosis) [8]. According to Klok et al., thrombotic events increase the risk of fatal outcome by 5.4 times [9]. In adults, abundant autopsy evidence suggests that thrombotic pulmonary microangiopathy is the underlying pathophysiological disorder in severe COVID-19 [10].

## 2. Methods

### 2.1. Study Design

A retrospective, pooled cohort study included data from the case histories of 60 patients with MIS, of whom 8 were diagnosed with thrombotic events.

### 2.2. Epidemiological Data

The association of thrombosis with a new COVID-19 infection was confirmed by evidence of a recent episode of acute infection with a positive PCR smear from the nasopharynx (1), detection of class G (7) and class M antibodies (3) to SARS-CoV-2, or histological detection of SARS-CoV-2 virus in tissues (1).

### 2.3. Diagnostics for SARS-CoV-2 Infection

SARS-CoV-2 infection was tested by polymerase chain reaction (PCR) to SARS-CoV-2 using nasopharyngeal swabs and enzyme immunoassay (ELISA) for G and M immunoglobulins and epidemiological history, including confirmed contact with a patient or carrier of SARS-CoV-2 in the family or community (kindergarten/school).

**Inclusion criteria were** compliance with WHO MIS-C diagnostic criteria and the presence of venous and arterial thrombosis confirmed by laboratory and radiological methods. Diagnosis of MIS-C was based on the WHO criteria (2020) [11]. The study inclusion period was November 2020 through November 2022.

### 2.4. Evaluated Parameters and Their Outcomes

*Demographic data:* Gender, age, time before admission.

*Laboratory tests:* All children were examined using standard laboratory methods (standard hematological and biochemical blood tests). Inflammatory parameters were reactive protein, procalcitonin, biomarkers of myocardial damage (troponin I, NT-proBNP), ferritin level, D-dimer, examination of hemostasis system parameters and markers of hereditary thrombophilia.

*Instrumental assessments:* All patients underwent standard electrocardiogram (ECG), Doppler echocardiography, multispiral computed tomography (CT) and/or magnetic resonance imaging (MRI) during ultrasound vascular scanning.

*Outcomes:* All patients were evaluated for disease outcomes (favorable outcome—discharge from hospital, unfavorable—lethal outcome), number of days spent in hospital, neurological сomplications (damage)—presence/absence of neurological symptoms, and recovery of lost functions. Radiology and ultrasound assessment of vessels examined permeability, presence/absence of stenoses and aneurysms, and presence/absence of thromboses.

### 2.5. Statistical Analysis

The sample size was not preliminarily calculated (pilot study). Data analysis was performed using STATISTICA 12 (StatSoft) and Microsoft Excel Professional Plus 2019. Quantitative variables were evaluated for their correspondence to the normal distribution using the Shapiro–Wilk criterion (for fewer than 50 subjects) or the Kolmogorov–Smirnov criterion (for more than 50 subjects), and a histogram with an approximating curve was also used. In the case of quantitative indicators with normal distribution, we calculated the arithmetic mean (M) and standard deviations (SD). Quantitative variables whose distribution differed from normal were described using median (Me) and lower and upper quartiles (25–75%). In order to study the relationship between the phenomena represented by quantitative data when the distribution differed from normal, we used a nonparametric method, calculation of the Spearman rank correlation coefficient (R). To study the relationship between quantitative variables and dichotomous variables we calculated the point-biserial Pearson’s correlation coefficient. The correlation coefficient was calculated for signs that had no missing data. Categorical data were described with absolute values and percentages. Category l data were compared using Pearson’s χ^2^ criterion or Fisher’s exact test, if expected frequency < 5. The ability of each variable to differentiate NBO from BO was evaluated with sensitivity and specificity analysis, AUC-ROC (area under receiver operating curve) with a 95% confidence interval (CI), calculating the odds ratio (OR) for the detection of the best cutoffs of continuous variables. The higher values of OR of variables indicated better discriminatory ability. Analysis of sensitivity and specificity was performed for each categorical variable. We avoided using the known “standard” threshold (i.e., a threshold previously reported in the literature or judged as clinically meaningful). We used the “best” thresholds obtained through the ROC curve analysis of our data because they provide the most appropriate means between sensitivity and specificity. Differences were considered statistically significant at *p* < 0.05.

## 3. Results

### 3.1. Analysis of Predictors of Thrombotic Events

During the COVID-19 pandemic (2020–2022), we observed 60 patients with COVID-19-associated MIS-C at the Regional Irkutsk Children’s Hospital from November 2020 to November 2022, among whom 8 (13.3%) had thrombotic complications, including one patient who was observed remotely. The children’s ages ranged from 4 months to 17 years. Of the eight patients, six (75%) were boys, for a male/female ratio of 3.0. Specifics of the clinical and laboratory signs of MIS-C in patients with thrombosis are shown in Table 1.

### 3.2. Course of Disease Characteristics of Patients with Thrombotic Events

Clinically manifest thrombosis in COVID-19-associated MIS-C was found in children from infancy to adolescence: four of the eight were under 5 years of age, including two infants < 1 year, and another four patients were between 8 and 17 years of age. The mean age (Me) was 7.5 years. The duration of the course of MIS-C before hospital admission was 1–6 days, and children being treated at home for “acute respiratory infections” during these days. Thrombotic events were documented immediately at the first admission to the clinic in six patients (one of six had thrombotic renal microangiopathy and further developed deep vein thrombosis and pulmonary embolism—PE—additionally). One more boy (aged 2 mo) was in the cardiac surgery department for ventricular septal defect at the time of the development of MIS-C with massive venous thrombosis. Another one boy (17 yo) developed deep vein thrombosis of the lower extremities and PE). In this way, three children developed thrombosis during their stays in the clinic, including PE in two, both with deep vein thrombosis of the lower extremities.

All eight patients developed thrombotic events after COVID-19-associated MIS-C symptoms (fever and high levels of inflammatory biomarkers), including seven who had signs of shock and multiorgan failure and were admitted immediately to the intensive care unit.

### 3.3. Clinical Manifestations

Patients with thrombosis associated with MIS-C did not differ from other MIS-C patients in terms of the main demographic and epidemiological data (Group 1 and Group 2, Table 1). Most of their clinical manifestations were also identical to other children with/without thrombosis, except for brain involvement (cerebral coma (*p* = 0.000006) or aseptic meningitis (OR 10.7 CI: 2.1–55.1) (Table 2). This is related to the above mentioned multiple cerebral vascular lesions due to arteritis (Table 3, case 1, 2, 3) or ischemic and hemorrhagic strokes (Table 3, case 1, 2), which were among the predominant thrombotic events in Group 1 patients. Secondary hemophagocytic syndrome occurred more frequently among patients with thrombotic complications. Children with thrombosis were more often admitted to the intensive care unit, primarily due to severe neurological involvement, associated with noncardiogenic pulmonary edema, and ischemic organ damage. The number of children with coronary dilatation was independent of thrombosis. No coronary thrombosis was identified.

### 3.4. Laboratory Data

There were no differences in biomarkers of inflammation and myocardial damage in patients with and without thrombosis. However, significant differences were found in platelet count and D-dimer levels. The average number of platelets in the first blood count on admission to the clinic in patients with thrombosis was 3.7 times lower than in children without thrombosis (76.8 × 10^9^/L vs. 284 × 10^9^/L), and 7 of 8 (87%) children with thrombosis had thrombocytopenia at the time of admission versus 21 of 52 (41%) of those without thrombosis (Table 1). Thrombocytopenia (≤103 × 10⁹/L) was associated with thrombosis (OR = 9.6, 95% CI: 1.1–83.3; *p* = 0.017) (Table 2). Mean D-dimer levels among patients with thrombosis were 2.3 times higher (6537 versus 2838 ng/mL, *p* < 0.0007). Increased D-dimer (>3778 ng/mL) was associated with the risk of thrombosis (OR = 22.8, 95% CI: 25–203.9) (Table 3). Coagulation parameters, such as PT, APPT and plasminogen, were initially within the reference values. In two of seven patients (28.6%), antithrombin III and protein C were reduced, including one patient in whom congenital antithrombin III deficiency was diagnosed by the detection of the *SERPINC1* heterozygous genetic variant.

High mean levels of NT-proBNP (5795 pg/mL; max—30 879 pg/mL) were recorded in five of seven patients (71.4%), indicating myocardial damage in patients with thrombosis. No coronary aneurysms, myocardial infarction, left ventricular dilatation or low ejection fraction were found. Increased NT-proBNP level (> 280 pg/mL) was associated with thrombosis (OR = 6.7, 95% CI: 1.15–39.1; *p* = 0.019) (Table 3).

### 3.5. General Characteristics of Patients with Thrombotic Events

The main features of the clinical data for patients, treatment and its results are shown in Table 2. In total, thrombosis occurred in 13.3% of children with MIS-C, including venous thrombosis in 10% and arterial thrombosis (including large artery thrombosis and secondary thrombotic microangiopathy) in 8.3%, with two patients having clinical manifestations of both venous and arterial thrombosis (Table 3, cases 3 and 4). Multiple thromboses (multiple localizations) occurred in half of the patients (4/8).

The main factors associated with thrombosis were coma, aseptic meningitis, platelets ≤ 103 × 10^9^/L, D-dimer > 3778 ng/mL, ferritin > 594 ng/mL, LDH > 382 U/L, and elevation of NT-proBNP (Table 2).

### 3.6. Venous Thromboses

Venous thromboses were observed in six patients, including four patients with deep vein thrombosis of the lower extremities (Table 3, patients 3, 4, 5, 7). Among these, two patients had pulmonary embolisms (3.3% of 60 patients). Patient 3, a 4-year-old girl, had disseminated PE after catheter-induced femoral vein thrombosis (Table 3). She immediately received intravenous bolus heparin and intravenous infusion was continued, after which PE signs stopped. In the second case (Table 3, patient 8), massive PE occurred in a 17-year-old boy with a severe premorbid background (Prader–Willi syndrome) after MIS-C therapy with intravenous dexamethasone at his local hospital, and the patient died.

In another 3-month-old child (Table 3, patient 8), hospital-acquired sepsis and multiple thrombosis of the superior vena cava and inferior vena cava branches and cerebral venous sinus thrombosis occurred during MIS-C. Treatment with anticoagulants, immunoglobulin and antibiotics resulted in the development of powerful multiple collaterals in the vena cava superior basin and recanalization of thrombus in the vena cava inferior. The patient recovered from all complications, and six months later closure of a ventricular septal defect was successfully performed.

Although all children were tested for genetic markers for thrombophilias, only one had a hereditary disorder of the hemostasis system (15-year-old boy, case 4). He was admitted to the clinic with DVT of the right lower extremity against the background of MIS-C, after which thrombosis of the femoral and iliac vein on the right side occurred. In this patient, the lower-extremity DVT was accompanied by a particularly deep and persistent decrease in antithrombin III (by 12–35%) even after fever receded, inflammatory biomarkers decreased, and clot recanalization occurred with restoration of blood flow in the affected vessels. A genetic study confirmed a heterozygous form of inherited antithrombin III deficiency in this patient. This thrombotic event was the first in his life, and after MIS-C with DVT occurred, heparin, followed by low-molecular-weight heparin treatment was prescribed with a subsequent switching to a continuous mode of warfarin treatment.

Among six children with venous thromboses, four survived with partial recanalization of the thrombus in the affected vessels and the development of powerful collaterals (Table 3, cases 4, 5, and 7). Another patient with a combination of venous thrombosis, renal thrombotic microangiopathy, and HPS (Table 3, case 3) was discharged with residual motor and cognitive defects, and one patient with background Prader-Willi syndrome died after developing PE.

### 3.7. Arterial Thromboses

Two patients had cerebral arterial thromboses (Table 3, cases 1 and 2). One of them (Table 3, case 1) also had additional arterial thrombosis of the extremities, which resulted in amputation of the right lower leg and osteonecrectomy of the terminal phalanges of two fingers of the right hand (Figure 1 and Figure 2). Histological examination of the amputated limb tissues revealed signs of necrotizing arteritis, thrombosis in arteries, capillaries and small veins (Figure 3), although there were no clinical manifestations of venous thrombosis. The presence of SARS-CoV-2 in vessel walls was also detected by RT-qPCR. This finding indicates the initiating role of the COVID-19 pathogen in triggering necrotizing arteritis, with distal gangrene accompanying arterial thrombosis of various diameters, including multiple capillaries. Molecular genetic testing for the CECR1 mutation to rule out DADA2 was negative.

Secondary thrombotic renal microangiopathy with hemolytic anemia phenomena occurred in three patients (cases 2, 3, and 6), including two as part of multiple thrombotic events, and in another patient as an isolated thrombotic event. These renal microthromboses related with COVID-19-associated multisystem inflammatory syndrome in one patient, were a trigger of atypical hemolytic uremic syndrome (case 3). This suspicion was further confirmed by molecular genetic testing, which revealed heterozygous CFHR1, CFHR3, and CFHR4 microdeletions. The presence of this microarray on chromosome 1 may be associated with the production of antibodies to factor H complement (CFH) and activation of the complement system through an alternative pathway with endothelial damage and, consequently, the development of thrombotic microangiopathy.

### 3.8. Treatment

Children with thrombosis in severe MIS-C (6 of 8) received pathogenetic therapy for multiorgan failure according to manifestations, including infusion therapy, artificial lung ventilation, diuretics, and inotropic drugs. All patients received immunomodulatory therapy, intravenous immunoglobulin and/or GCS (dexamethasone, methylprednisolone) with no significant differences in the frequency of use in the group with and without thrombosis (Table 1). Intravenous immunoglobulin was administered in a single dose of 1.5–2.0 g/kg/course. Dexamethasone was administered intravenously at a dose of 10–20 mg/m^2^, followed by tapering over 3–4 weeks. Combined therapy with intravenous immunoglobulin and GCS was used in 6 (75%) patients with thrombosis and in 23 (44%) of the remaining patients.

Due to resistance to treatment with intravenous immunoglobulin and GCS, two of the eight patients with thrombotic events were also treated with biologics. In one patient, tocilizumab 8 mg/kg (single administration) and eculizumab were successfully used. After tocilizumab administration, it was possible to interrupt the long refractory fever. After introduction of eculizumab, which is an inhibitor of complement fraction C5, hemolytic crises associated with aHUS stopped. In another patient undergoing refractory fever and persistent elevation of inflammatory biomarkers even after treatment with 2.0 g/kg/course of IVIG, dexamethasone and pulse therapy with methylprednisolone and cyclophosphamide, followed by etanercept, led to suppression of clinical and laboratory activity.

In the group without thrombosis, biologic drugs were administered less frequently (2 of 52 patients; in one case tocilizumab, in the other, anakinra). Thus, in view of the tendency toward a higher frequency of use of biologics in group 1, thrombosis can be considered an indicator of a higher degree of severity of MIS-C, although the difference in the frequency of use between groups is not statistically significant, due to the small number of groups (Table 1)

At first, intravenous infusion of heparin was carried out in the ICU department with a starting dose of 10–20 units/kg/h, with titration of the dose under control of the APPT lengthening by 1.5–2 times every six hours and subsequent transition to low-molecular-weight heparin (nadroparin). After recanalization, acetylsalicylic acid was added for three months for thrombosis prophylaxis. The average length of hospital stay for children with thrombosis was higher than for patients without thrombosis, which is associated with the severity of organ dysfunction due both to MIS-C and the thrombotic events.

### 3.9. Outcomes

Arterial thromboses in patients primarily involved the cerebral vessels (in three of eight patients, 25%) and the renal capillary vessels (in three of eight, 37.5%). Cerebral artery thrombosis led to devastating consequences in the form of extensive ischemic and hemorrhagic strokes, resulting in leukomalacia and cerebral dropsy with severe residual neurological deficits. Interestingly, all arterial thromboses occurred in the two patients with MIS-C who had signs of Kawasaki syndrome. According to follow-up data, one of these patients had an almost complete recovery of cognitive and motor functions after 2 years.

Venous thrombosis in the remaining patients regressed without severe residual sequalae. Secondary thrombotic microangiopathy in one case out of three was resolved with eculizumab therapy, and in two more patients, the recovery was complete without the use of biologics.

All patients with thrombotic events (8) survived, except one who died from pulmonary embolism (12.5%) with a background of Prader-Willi syndrome. In addition, three of seven patients were discharged with severe disabling damage of the nervous system due to thrombotic complications of MIS-C (consequences of extensive multiple strokes—2, signs of cerebral vasculitis and leukomalacia—1). Two of these three patients had excellent recovery of cognitive and motor functions after 2 years of observation after discharge as a result of rehabilitation, but one has not had any significant recovery.

The remaining patients, both with and without thrombosis, feel well, continue to be monitored without complications, and are leading a normal lifestyle.

In further follow-up of patients with thrombotic events that occurred during MIS-C over 1–2.5 years, none had new thromboses, coronary aneurysms or myocardial infarctions. Coronary dilation, noted in 12.5% of all 60 patients, disappeared during this observation period. No other serious disease (including rheumatic disease) has been observed in any of the patients. Dynamic laboratory tests (hematological tests, including biomarkers and tests of hemostasis system indicators) have not revealed any deviations from the norm.

## 4. Discussion

Our study shows the variety of thrombotic events occurring in patients with MIS-C. In this regard, close attention to the risks and manifestations of thrombotic complications is required, based on the understanding of the specific pathogenetic features of the new coronavirus infection. Immunothrombosis is a condition that occurs in many infections (primarily sepsis) in which the immune system and the clotting system interact to block antigen and its spread [12]. Immunothrombosis is induced by neutrophils and monocytes, which release tissue factor and extracellular nucleosomes (NETosis) and destroy endogenous anticoagulants, activating the coagulation induced by inflammation [13,14]. The formation of neutrophil extracellular traps (NETs) occurs when the nuclear envelope and granular membranes are disintegrated and a network of interwoven DNA strands mixed with histone proteins and various antimicrobial molecules from cellular granules are ejected from the cell, eventually leading to extracellular killing of microorganisms.

COVID-19 infection has been found to trigger many pathogenetic pathways leading to thrombosis, confirming the concept of COVID-19-associated coagulopathy, which makes it possible to isolate various biomarkers indicating the development of thrombosis [15]. Although its main cause is thought to be inflammation of the vascular wall, thrombosis is not only due to endothelial damage. There are data about the role of SARS-CoV-2-activated platelets in the pathogenesis of thrombosis in COVID-19 [16]. Megakaryocytes actively capture SARS-CoV-2 viruses and affect their transcriptome, causing changes in the phenotype and functional ability of the platelets [16]. SARS-CoV-2 virions have been found in megakaryocytes and blood plates of bone marrow, lung tissue and blood of adult patients with COVID-19 [16]. Thus, platelets may also deliver viral particles to the foci of inflammation of the vascular wall.

It was found that platelets produced by megakaryocytes with an altered genome under the influence of SARS-CoV-2 are less mature and have a higher prothrombotic status: they agglomerate more easily and quickly with various agonists, including ADP, adrenaline, and collagen, have more active oxidative phosphorylation and glycolysis, and are enriched in genes responsible for degranulation. These immature platelets are larger in diameter, contain more RNA, and can produce various proteins typical of platelet activation (GP2b/IIIa, P-selectin). Changes in blood plate number, size, and maturity are associated with disease severity and all-cause mortality among patients hospitalized with COVID-19 [17].

In contrast to adults, there are few publications on thrombosis in children associated with new coronavirus infection induced by SARS-CoV-2. They have mostly been described in patients with multisystem inflammatory syndrome, and less often in patients with acute COVID-19 bronchopulmonary infection. In some cases, thrombosis was found in asymptomatic forms of the disease [18,19,20]. In pediatric multisystem inflammatory syndrome, the laboratory findings suggest a procoagulant status with prominent increase of the D-dimer level, similar to patients with cytokine storm in acute COVID-19 infections [19,20,21]. Venous thrombosis is predominantly described, while reports of arterial thrombosis in children with COVID-19 are scarce. [22] A recent study by Whitworth et al. [18] found that the incidence of thrombosis in multisystem inflammatory syndrome in children was 6.5%, and was highest compared with acute SARS-CoV-2 infection (2.1%) and its asymptomatic form (0.7%). The incidence of thrombosis in our group of children with MIS-C was twice as high, at 13.3%; this is probably due to the fact that we included secondary thrombotic microangiopathy, which is usually described separately in the literature [23,24]. Both venous and arterial thromboses occurred in our MIS-C cohort, and no hemorrhages were observed. Seven of eight patients were treated in the intensive care unit, including five suffering shock and multiple organ failure and one patient with bilateral deep vein thrombosis of the lower limbs and IVC due to the threat of PE, which was prevented. Another patient with unilateral deep vein thrombosis of the lower extremity was treated in the general ward.

Deep vein thrombosis of the lower extremities was prevalent among all thrombotic events, which is also typical for COVID-19 in adults. However, unlike adults, for whom arterial thrombosis is a rare event [25], it was recorded in four of eight of our patients.

COVID-associated coagulopathy (CAC), which is characterized by increased D-dimer levels in the absence of marked shifts in other standard coagulation indices, is considered to be the cause of thrombotic events in COVID-19 and therefore differs from classical disseminated intravascular coagulation (DIC), but can progress to it [26,27]. This was the case for our patients (Table 3). There were no signs of obvious DIC syndrome in the coagulopathy of consumption stage and no bleeding in any of the patients.

Direct exposure of endotheliocytes to SARS-CoV-2 and immune system activation with the development of vascular wall inflammation and immunothrombosis have been found to result in CAC. As a result, micro- and macrovascular thrombotic events involve arterial and capillary vessels as well as venules and large veins [28]. According to our observations, such pathophysiological mechanisms in children with multisystem inflammatory syndrome associated with COVID-19 are confirmed by both clinical and pathomorphological data. We observed large arterial thrombosis in two patients. In one case, arterial thrombosis of the extremities was combined with severe brain damage due to multiple foci of hemorrhagic and ischemic strokes (Table 3, case 1), and in the second case (Table 3, case 2), there was a bilateral stroke. This localization of thrombosis determined the most severe prognosis for patients in our observation group; these patients survived with severe neurological sequalae.

Patient 3 had clinical manifestations of both venous and arterial thrombosis, and CT brain angiography showed changes typical of arteritis (arterial vessels with lumen disruption).

Among eight patients, arterial thrombosis (large vessel and capillary, i.e., thrombotic renal microangiopathy) developed only in those four children who had a severe course of MIS-C with signs of Kawasaki syndrome, shock, and multiple organ failure (cases 1, 2, 3, and 6). Therefore, it is obvious that in addition to the clinical manifestations of Kawasaki syndrome, the development of arterial thrombosis is an indicator of SARS-CoV-2-induced vasculitis. COVID-associated vasculopathy occurred among our patients both with extremely dramatic manifestations in the form of cerebral vascular thrombosis (cases 1 and 2, Table 3), including in combination with distal gangrene, and with less severe observations of thrombotic renal microangiopathy (cases 2, 3, 6).

Secondary renal thrombotic microangiopathy is a group of disorders manifested by microangiopathic hemolytic anemia, thrombocytopenia, and capillary thrombosis in various organs, leading to ischemic organ damage. The pathogenetic mechanism of renal thrombotic microangiopathy is associated with viral infection and includes (a) direct damage to the endothelium, (b) the presence of acquired disintegrin and metalloproteinase inhibitors involving thrombospondin type 1 (ADATS13 family protein), (c) the presence of lupus anticoagulant, and (d) uncontrolled complement system activation [29]. The alternative pathway of complement system activation was found to create a predisposition to the development of COVID-19-associated renal thrombotic microangiopathy. In particular, a prospective cohort observational study identified several abnormal variants of complement factor H, C5b-9 (ADAMTS-13) in patients with COVID-19 who developed atypical hemolytic syndrome [30]. In our group, atypical hemolytic uremic syndrome (aHUS) was diagnosed in a 4-year-old girl (case 3) along with cerebral vasculitis and pulmonary embolism, with no ADAMTS-13 factor detected. The C3–C4 complement fractions were at the lower end of the normal range, but heterozygous CFHR1, CFHR3, and CFHR4 microdeletions associated with the production of complement factor H (CFH) antibodies that enhance complement activity were found. Eculizumab, a C5b convertase inhibitor, was successfully used in the treatment in addition to dexamethasone, IVIG, and tocilizumab. There are also descriptions in the literature of effective treatment with eculizumab of cases of atypical HUS in COVID-19 in both adults and children [31,32].

The authors did not find coronary thrombosis or coronary aneurysms in patients, only moderate coronary dilatation in 13.3% of the total number of patients, which is consistent with the literature [33,34,35,36]. The absence of myocardial infarction/giant coronary aneurysms with a simultaneous increase in troponin and NT-proBNP levels suggests the presence of coronary thrombotic microangiopathy, which requires further investigation.

## 5. Research Limitations

Our study has several limitations, including the small number of patients studied, the retrospective nature of the study, and the partially incomplete examination data for the children. The lack of a unified diagnostic and therapeutic algorithm may also have influenced the results of the study. It is unclear whether time to admission, timeliness of diagnosis, or therapeutic tactics influenced the incidence of thrombosis and outcomes.

## 6. Conclusions

Thrombotic events are associated with MIS-C. The main predictors of thrombosis were increased D-dimer, thrombocytopenia, ICU admission, and noncardiogenic pulmonary edema. Deep vein thrombosis, pulmonary embolism, cerebral artery thrombosis and thrombotic renal microangiopathy were the most common thrombotic events. Moderate transient coronary artery dilatation was noted, but coronary aneurysms, coronary thromboses and myocardial infarctions were not found in any patient. Cerebral arterial thromboses, in combination with Kawasaki-like symptoms and shock, were the most serious complications in our group of MIS-c patients with thrombotic events. We propose including antibodies to SARS-CoV-2 in the algorithm for diagnosing all childhood thromboses along with other known thrombotic factors.

## Figures and Tables

**Figure 1 biomedicines-11-02206-f001:**
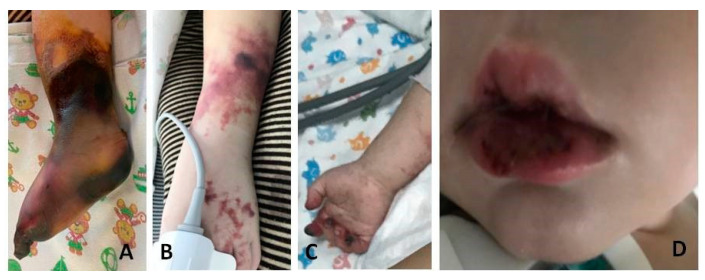
Gangrene of the right lower limb (**A**), rashes with superficial necrosis on the left leg (**B**), necrosis of the distal phalanges of the right hand (**C**), cheilitis (**D**) in patient 1 with multisystem inflammatory syndrome associated with COVID-19.

**Figure 2 biomedicines-11-02206-f002:**
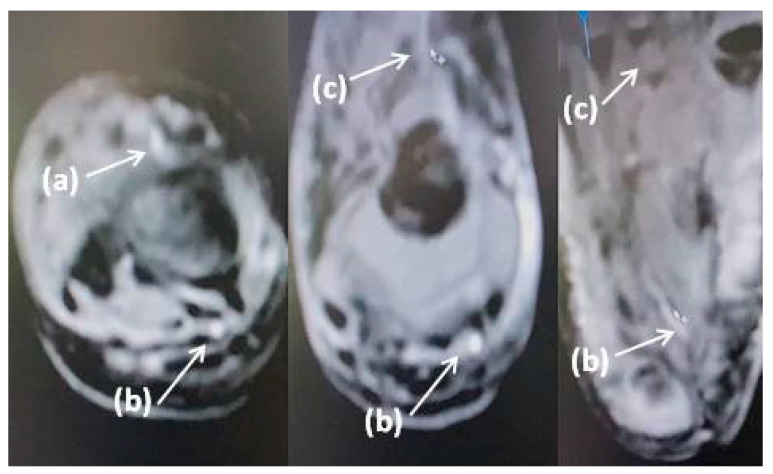
MRI of the right lower leg and foot: (**a**) anterior tibial artery and branches, (**b**) posterior tibial artery and branches, (**c**) loss of visualization of the anterior tibial artery and branches (patient 1).

**Figure 3 biomedicines-11-02206-f003:**
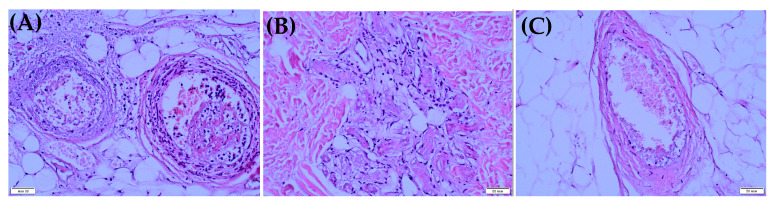
Morphological examination of the removed tissues of patient 1. (**A**) Cross section of two arteries in adipose tissue, with pronounced neutrophil infiltration and hemorrhages in the wall and mixed-structure thrombi in the lumen (mixture of erythrocytes and fibrin). The wall of the right artery is almost destroyed, with distinct leukoclasia (nuclear dust in the form of small dark dots). (**B**) A fragment of dermal tissue with the presence of capillaries. There are fibrin clots in the lumen and reactive endothelial changes in the form of polymorphism cells and hyperchromia of the nuclei. (**C**) Vein in adipose tissue has thrombus of mixed structure in the lumen, but without signs of phlebitis.

**Table 1 biomedicines-11-02206-t001:** Comparison of clinical and laboratory data in patients with MIS-C complicated by thrombosis.

Characteristics	MIS-C with Thrombosis (n = 8)	MIS-C without Thrombosis (n = 52)	*p*
**Demographics**			
Age (months), mean (min–max)	90.3 (3–204)	88.1 (3–204)	0.757
Gender, boys, n (%)	6 (75)	28 (54)	
Number of days before admission, mean (min–max)	6.8 (2–15)	10.4 (2–32)	0.096
Duration of hospitalization, days, mean (min–max)	59 (27–90)	31 (10–80)	0.0086
**Epidemiological data**			
Number of children with detected IgG to SARS-CoV-2, n (%)	6 (86)	48/57 (96)	0.253
Number of children with detected IgM to SARS-CoV-2, n (%)	0 (0)	10/58 (17)	0.197
Family contact, n (%)	2 (25)	28 (55)	0.115
Clinical symptoms			
Duration of fever (days), mean (min–max)	19 (1–59)	12 (3–32)	0.388
Gastrointestinal symptoms, n (%)	4 (50)	35 (67)	0.339
Neurological symptoms, n (%)	6 (75)	24 (46)	0.128
Coma, n (%)	3 (37)	0 (0)	0.000006
Aseptic meningitis, n (%)	5 (62)	7 (13)	0.0012
Polyserositis, n (%)	4 (50)	30 (58)	0.682
Pharyngitis, n (%)	2 (25)	29 (56)	0.104
Cervical lymphoadenopathy, n (%)	7 (87)	45 (86)	0.940
Rash, n (%)	2 (25)	32 (61)	0.052
Conjunctivitis, n (%)	4 (50)	38 (73)	0.184
Cheilitis n (%)	3 (37)	37 (71)	0.060
Glossitis/stomatitis, n (%)	5 (62)	39 (75)	0.456
Respiratory symptoms, n (%)	3 (37)	13 (25)	0.456
Palms and feet edema/desquamation, n (%)	5 (62)	39 (75)	0.456
Soft tissues sedema, n (%)	5 (62)	38 (73)	0.536
Hepatomegaly, n (%)	6 (75)	41 (79)	0.805
Splenomegaly, n (%)	4 (50)	24 (46)	0.839
Arthritis/arthralgia, n (%)	1 (12)	21 (40)	0.127
Myocardial damage, n (%)	5 (62)	33 (64)	0.903
Acute heart failure, n (%)	3 (37)	9 (17)	0.183
Pulmonary edema, n (%)	4 (50)	9 (17)	0.036
Coronary artery dilatation, n (%)	1/8 (12.5)	7 (13.5)	0.952
Myocarditis, n (%)	5 (62)	25 (48)	0.447
Pericarditis, n (%)	2/7 (28)	26 (50)	0.286
ICU admission, n (%)	7 (87)	24 (46)	0.029
Artificial lung ventilation, n (%)	3 (37)	7 (13)	0.089
Macrophage activation syndrome, n (%)	4 (50)	9 (17)	0.036
**Laboratory data**			
Hemoglobin (n.v. 120–140 g/L), mean (min–max)	93 (73–114)	104 (63–158)	0.111
White blood cells (n.v. 4–9 × 10^9^/L), mean (min–max)	22.4 (6.6–58)	13.9 (2–43)	0.126
Patients with leukocytosis, n (%)	3 (37)	19 (36)	0.958
Platelets (n.v. 150–350 × 10^9^/L), mean (min–max)	76.8 (4–182)	284 (26–1077)	0.0132
Patients with thrombocytosis, n (%)	0 (0)	15 (28)	0.079
Patients with thrombocytopenia, n (%)	7 (87)	29 (56)	0.088
Patients with platelets < 103 × 10^9^/L *, n (%)	7 (87)	21 (41)	0.014
ESR (n.v. 0–30 mm/h), mean (min–max)	39 (4–70)	42 (2–112)	0.856
Patients with increased ESR, n (%)	6 (75)	37 (71)	0.822
C-reactive protein (n.v. 0–5 mg/L), mean (min–max)	107.2 (19.7–245.6)	124 (0.3–335)	0.940
Ferritin (n.v. 7–140 ng/mL), mean (min–max)	583 (221–853)	454 (43–2254)	0.106
Patients with increased ferritin, n (%)	6/6 (100)	33/46 (72)	0.132
ALT (n.v. <33 IU/L), mean (min–max)	658 (9.9–3101)	135 (10–2816)	0.320
Patients with increased ALT, n (%)	5 (62)	21/51 (41)	0.258
AST (n.v. <39 IU/L), mean (min–max)	608 (12.6–2383)	128 (13–2402)	0.413
Patients with increased AST, n (%)	5 (62)	30/51 (59)	0.843
Total bilirubin (n.v. 0–21 μmol/L), mean (min–max)	11.1 (1.5–28.8)	17.6 (1.2–372)	0.887
Albumin (n.v. 28–54 g/L), mean (min–max)	31.6 (25–42.3)	32.2 (20–46.1)	0.760
Triglycerides (n.v. 0.2–1.7 mmol/L), mean (min–max)	3.1 (0.7–7.3)	2.5 (0.6–7.7)	0.732
Creatinine (n.v. 28–70 μmol/L), mean (min–max)	80.1 (30–225)	69.1 (14–298)	0.769
LDH (n.v. 240–480 IU/L), mean (min–max)	1020 (148–3325)	410.3 (79–2390)	0.063
Patients with increased LDH, n (%)	4 (50)	11/51 (21.57)	0.085
Procalcitonin (n.v. 0–0.5 μg/mL), mean (min–max)	53.9 (0.1–183)	10.8 (0.1–88.6)	0.458
Аntithrombin III (n.v. 80–120%), mean (min–max)	69.8 (16.6–103.7)	nd	nd
Protein C (n.v. 60–140%), mean (min–max)	96.0 (28.7–178.1)	nd	nd
Troponin (n.v. 0–14 ng/mL), mean (min–max)	16.3 (0.1–100)	23.3 (0–450)	0.789
Fibrinogen (n.v. 2.0–3.9 g/L) mean (min–max)	2.7 (0.1–4.1)	3.8 (0.1, 8.8)	0.191
D-dimer (n.v. 0–500 ng/mL), mean (min–max)	6537 (3026–14,800)	2838 (433–8836)	0.0007
Patients with D-dimer > 3778 ng/mL *, n (%)	7 (87)	12/51 (23)	0.0003
NT-proBNP (n.v. < 160 pg/mL *), mean (min–max)	5795 (10–30,879)	596 (10–6783)	0.066
Increase in NT-proBNP, n (%)	5/7 (71)	16/48 (33)	0.053
Creatine kinase-MB (n.v. 0–25 IU/L), mean (min–max)	31.9 (9–70.9)	27.4 (5.0–94.7)	1.0
**Treatment and outcome**			
IVIG (1.8–2 g/kg/course), n (%)	6 (75)	32 (61)	0.462
Acetylsalicylic acid, n (%)	4 (50)	33 (63)	0.466
GCS (dexamethasone 10–20 mg/m^2^/day), n (%)	6 (75)	35 (67)	0.663
Biologic treatment, n (%)	2 (25)	3 (6)	0.066
Deaths, n (%)	1 (13)	2 (4)	
CNS damage (stroke), n (%)	3 (38)	0	0.083

* Calculated with AUC-ROC analysis; Abbreviations (in alphabetical order): ALT—alanine aminotransferase, AST—aspartate aminotransferase, CK-MB–creatine kinase-MB, CNS—central nervous system, ESR—erythrocyte sedimentation rate, GCS—glucocorticosteroids, ICU—intensive care unit, IVIG—intravenous immunoglobulin, LDH—lactate dehydrogenase, n.d.—no data, n.v.—normal value, PCT—procalcitonin test, NT-proBNP—N-terminal brain natriuretic peptide.

**Table 2 biomedicines-11-02206-t002:** Factors, associated with thrombosis in patients with MIS-C.

Predictors	OR (95% CI)	*p*
Coma	-	0.000006
Aseptic meningitis	10.7 (2.1–55.1)	0.001
Platelets ≤ 103 × 10^9^/L	9.6 (1.1–83.3)	0.017
D-dimer > 3778 ng/mL	22.8 (25–203.9)	0.0003
Ferritin > 594 ng/mL	8.7 (1.4–54.9)	0.010
LDH > 382 U/L	6.2 (1.13–33.9)	0.022
Elevation of NT-proBNP levels	5.0 (0.9–28.7)	0.053
Elevation of NT-proBNP > 280 pg/mL	6.73 (1.15–39.1)	0.019

Abbreviations (in alphabetical order): CI—confidence interval, LDH—lactate dehydrogenase, OR—odds ratio, NT-proBNP—N-terminal brain natriuretic peptide.

**Table 3 biomedicines-11-02206-t003:** Patient characteristics.

ID, #	Sex, Age	SARS-CoV-2 Confirmation	Clinic Signs	Type of Thrombosis	Treatment of MIS-C/Thrombosis	Outcome
1	M. 1.5 y.	IgG (+)	MIS-C, Kawasaki-like features, MOFS, shock, myocarditis, severe CNS damage, distal gangrene	Arterial thrombosis of the lower extremities, multiple ischemic strokes, venous thrombosis (morphological)	IVIG 2.0 g/kg 2 courses, DEXA 15 mg/m^2^/day, MP pulse therapy, cyclophospha-mid pulse therapy, etanercept, heparin/nadroparin, ASA	Recovery with residual severe neurological sequelae
2	M. 8 y.	IgG (+)	MIS-C, Kawasaki-like features, mild coronary dilation, MOFS, shock, severe CNS involvement,	Bilateral ischemic stroke Secondary renal TMA	IVIG 2.0 g/kg/course DEXA 15 mg/m^2^/day heparin/nadroparin, ASA	Full recovery in 2 years
3	F. 4 y.	IgG (+)	Kawasaki-like features, MIS-C, shock, MOFS, severe damage to the central nervous system, aHUS, secondary HFS	Secondary renal TMA (aHUS), catheter-associated thrombosis of the femoral vein, disseminated PE	IVIG 2 courses, DEXA15 mg/m^2^/day Tocilizumab, Eculizumab, Heparin/nadroparin	Recovery with residual neurological sequelae in 2 years and full restoration of the renal function on the eculizumab
4	M. 15 y.	IgG (+)	MIS-C, (hereditary AT III deficiency)	Deep vein thrombosis of the upper and lower extremities	IVIG, dexamethasone, ASA, Heparin/nadroparin/warfarin	Recanalization of clots, recovery
5.	F. 13 y.	IgG (+)	MIS-C	Deep vein thrombosis of the lower extremity	Heparin/nadroparin/warfarin	Recanalization of clots, recovery
6	M. 5 m.	IgG (+)	MIS-C with Kawasaki-like features, severe, MOFS, shock	Secondary renal TMA	IVIG, DEXA, heparin/nadroparin, ASA	Full recovery
7	M. 3 m.	PCR (+) IgG (+)	MIS-C, severe MOFS, shock, RDS, myocarditis, sepsis, severe CNS involvement, (background—CVSD)	Cerebral sinus thrombosis, deep vein thrombosis—branches of VCS, VCI	IVIG, DEXA, Heparin/nadroparin	Recanalization of clots, recovery
8	M. 17 m	IgG (+)	MIS-C, severe MOFS, secondary HFS. Comorbidity—Prader–Willi syndrome	Deep vein thrombosis, PE	DEXA, heparin/nadroparin	Fatal outcome

Abbreviations (in alphabetical order): ASA—acetylsalicylic acid, AT III—antithrombin III, CNS—central nervous system, CVSD—congenital ventricular septal defect, DEXA—dexamethasone, HFS—hemophagocytic syndrome, HUS—hemolytic uremic syndrome, IVIG—intravenous immunoglobulin, MAHA—microangiopathic hemolytic anemia, MIS-C—multisystem inflammatory syndrome in children, MOFS—multiple organ failure syndrome, PE—pulmonary embolism, RDS—respiratory distress syndrome, TMA—thrombotic microangiopathy, VCI—vena cava inferior, VCS—vena cava superior.

## Data Availability

The data presented in this study are available on reasonable request from the corresponding author.

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
