# Peer review of "Thrombosis in Multisystem Inflammatory Syndrome Associated with COVID-19 in Children: Retrospective Cohort Study Analysis and Review of the Literature"

_biomedicines, 2023, doi:10.3390/biomedicines11082206_

Round 1

Reviewer 1 Report

A very interesting manuscript summarizing the course of COVID-19 in children.
I have a few comments to work with:
- "proBNP" appears in the text, and BNP/proBNP material/methods. Basically, two variants of the assays NT-proBNP (currently recommended especially in monitoring the course of heart failure) and BNP, otherwise known as proBNP, are used. Please specify clearly which assay was used .
- in the work, for the term 'micro', eg in micromol, the Greek letter micro appears and sometimes mc. Please check the text and use Greek letters everywhere.
- ASA was used and not heparin (UFH/LMWH)? Why, please explain. In the treatment of thrombosis, a very long intravenous infusion (I understand UFH) was used under the control of aPTT. How often were blood samples taken every 6 hours more? have there been HIT incidents? Why LMWH was not used from the beginning - please discuss these issues.
- have DOACs been used - e.g. rivaroxaban registered for use in children?
- what was the activity of antithrombin III? There is information about the lowering in the text, but it is not visible in the data table.

Author Response

Dear Reviewer!

Thank you so much for your positive evaluation of our manuscript. Our answers (A) on your queries (Q) are below and highlighted by color in the manuscript.

I have a few comments to work with:
Q1) - "proBNP" appears in the text, and BNP/proBNP material/methods. Basically, two variants of the assays NT-proBNP (currently recommended especially in monitoring the course of heart failure) and BNP, otherwise known as proBNP, are used. Please specify clearly which assay was used.

A1) Dear Reviewer! We used NT-proBNP. The correction was made everywhere in the manuscript

Q2) - in the work, for the term 'micro', eg in micromol, the Greek letter micro appears and sometimes mc. Please check the text and use Greek letters everywhere.

A2) Dear Reviewer! Everywhere we have used now Greek symbol μ

Q3) - ASA was used and not heparin (UFH/LMWH)? Why, please explain. In the treatment of thrombosis, a very long intravenous infusion (I understand UFH) was used under the control of aPTT. How often were blood samples taken every 6 hours more? have there been HIT incidents? Why LMWH was not used from the beginning - please discuss these issues.

A3) Dear Reviewer! We used UFH in the acute phase in the form of the long IV infusion with close monitoring of aPTT in the ICU department with the following switching to LMWH. Fortunately no cases of HIT were observed. The UFH used for quick hypocoagulation. ASA was used as additional therapy in patients being on the recalanisation stage and after discharge from the hospital for new thrombosis prevention. Because all patients were with MIS-C the ASA was also used as a part of MIS-C treatment. During the peak of pandemy there was also situation of the lack of access to LMWH due to disease outbreak. The information in the manuscript has been updated.
Q4) - have DOACs been used - e.g. rivaroxaban registered for use in children?

A4) Dear Reviewer!  In some patients we used warfarin, but it is difficult to monitoring and dose titration. In some patients we used rivaroxaban. It was not approved in children in our country, but local temporarily recommendations for COVID-19 treatment allowed to used medications off-label.

Q5- what was the activity of antithrombin III? There is information about the lowering in the text, but it is not visible in the data table.

A5. Dear Reviewer! The antithrombin III level was measured in seven patients with thrombosis. The data about antithrombin III and Protein C added to the table and some more information added in the results.

Dear Reviewer! I hope the manuscript became better after your suggestions and recommendations.

On behalf of the Authors

Mikhail Kostik, MD, PhD, Professor

Reviewer 2 Report

The study commences by highlighting the unique properties of the SARS-CoV-2 virus in causing hyperinflammatory syndrome and cytokine storm, as well as widespread endotheliitis and thrombotic microangiopathy, particularly prominent in adult patients who succumbed to severe forms of the disease. Notably, the authors underscore the scarcity of reports concerning thrombotic events in children with COVID-19, making this research all the more indispensable.

The findings present a cohort of 60 pediatric patients with MISc, of whom 8 (13.3%) were diagnosed with venous and/or arterial thrombosis, confirmed through laboratory and ultrasound and/or X-ray methods. The age distribution and gender ratio of the affected children reveal interesting patterns, providing a basis for future investigations.

One of the most significant outcomes of the study is the identification of various thrombotic manifestations in pediatric patients. Venous thrombosis, involving both deep veins of the lower extremities and pulmonary embolism, were the most commonly observed. Additionally, cases of cerebral venous sinus thrombosis and thrombosis of the branches of the vena cava underscore the diverse nature of thrombotic complications in children with MISc.

Moreover, the study highlights the occurrence of extensive bilateral stroke resulting from thrombosis of large cerebral arteries, a phenomenon previously infrequently reported in the context of COVID-19 in children. This insight not only advances our understanding of the disease but also underscores the importance of vigilance in identifying and managing such complications.

The clinical characteristics associated with thrombosis, such as elevated D-dimer levels, thrombocytopenia, increased proBNP, cerebral coma, and aseptic meningitis, provide a valuable framework for early identification and intervention. The authors report the application of a range of therapeutic modalities, including immunomodulatory therapy, anticoagulant therapy, and biologics, in managing thrombotic events, suggesting promising avenues for future research and treatment strategies.

Furthermore, the study emphasizes that while thrombotic complications in children with MISc can be severe, the mortality rate remains relatively low, with only one death reported among the eight patients. Understanding these outcomes contributes to the establishment of more effective and targeted interventions to reduce morbidity and mortality in this specific patient population.

In conclusion, this study represents a pioneering effort in uncovering the unique thrombotic properties of SARS-CoV-2 in pediatric patients with MISc. By elucidating the main predictors of thrombosis and outlining the various manifestations, the research enables healthcare providers worldwide to better comprehend and address the challenges posed by thrombotic events in this vulnerable population. Undoubtedly, this study will have a profound impact on the development of evidence-based therapeutic strategies, ultimately leading to improved outcomes for children affected by COVID-19-related thrombosis.

However, I believe that the bibliography needs to be supplemented and the authors should also refer to the following articles in their introduction and discussion:

doi: 10.1016/j.jtha.2023.05.020.

doi: 10.1016/j.cpcardiol.2022.101186.

doi: 10.5603/DEMJ.a2022.0039

doi: 10.5603/CJ.a2022.0123

doi: 10.1016/j.bcmd.2023.102746. 

doi: 10.19204/2023/thpt4

doi: 10.5603/DEMJ.a2023.0007

doi: 10.1097/MPH.0000000000002590.

doi: 10.1016/j.jtha.2023.05.020. 

doi:  10.19204/2023/acmp3

doi: 10.5830/CVJA-2022-025.

doi: 10.5603/DEMJ.a2021.0024

doi:  10.19204/2022/SSCT4

doi: 10.20452/pamw.15685. 

doi: 10.1002/rmv.2432.

doi: 10.1002/iid3.838.

doi: 10.1016/j.arcped.2023.01.006.

Author Response

Dear Reviewer!

Thank you so much for your very positive mark of our manuscript.

The recommended manuscripts were added to the discussion and to the reference list. All changes were highlighted with color. I hope the manuscript became better after your suggestions and recommendations.

On behalf of the Authors

Mikhail Kostik, MD, PhD, Professor

Round 2

Reviewer 1 Report

The authors have introduced corrections, clarified unclear wording, thus I recommend the manuscript for publication.